# Highly Efficient Flexible Roll-to-Roll Organic Photovoltaics Based on Non-Fullerene Acceptors

**DOI:** 10.3390/polym15194005

**Published:** 2023-10-06

**Authors:** Yu-Ching Huang, Hou-Chin Cha, Shih-Han Huang, Chia-Feng Li, Svette Reina Merden Santiago, Cheng-Si Tsao

**Affiliations:** 1Department of Materials Engineering, Ming Chi University of Technology, New Taipei City 24301, Taiwan; 2Organic Electronics Research Center, Ming Chi University of Technology, New Taipei City 24301, Taiwan; 3Department of Chemical and Materials Engineering, College of Engineering, Chang Gung University, Taoyuan 33302, Taiwan; 4Institute of Nuclear Energy Research, Taoyuan 32546, Taiwan; 5Department of Materials Science and Engineering, National Taiwan University, Taipei 10617, Taiwan

**Keywords:** organic photovoltaic, slot-die, roll-to-roll, flexible, non-fullerene

## Abstract

The ability of organic photovoltaics (OPVs) to be deposited on flexible substrates by roll-to-roll (R2R) processes is highly attractive for rapid mass production. Many research teams have demonstrated the great potential of flexible OPVs. However, the fabrication of R2R-coated OPVs is quite limited. There is still a performance gap between the R2R flexible OPVs and the rigid OPVs. In this study, we demonstrate the promising photovoltaic characteristics of flexible OPVs fabricated from blends of low bandgap polymer poly[(2,6-(4,8-bis(5-(2-ethylhexyl)thiophen-2-yl)-benzo[1,2-b:4,5-b′]dithiophene))-alt-(5,5-(1′,3′-di-2-thienyl-5′,7′-bis(2-ethylhexyl)benzo[1′,2′-c:4′,5′-c′]dithiophene-4,8-dione)] (PBDB-T) and non-fullerene 3,9-bis(2-methylene-(3-(1,1-dicyanomethylene)-indanone))-5,5,11,11-tetrakis(4-hexylphenyl)-dithieno[2,3-d:2′,3′-d′]-s-indaceno[1,2-b:5,6-b′]dithiophene (ITIC). We successfully R2R slot-die coated the flexible OPVs with high power conversion efficiency (PCE) of over 8.9% under irradiation of simulated sunlight. Our results indicate that the processing parameters significantly affect the PCE of R2R flexible OPVs. By adjusting the amount of solvent additive and processing temperature, as well as optimizing thermal annealing conditions, the high PCE of R2R slot-die coated OPVs can be obtained. These results provide significant insights into the fundamentals of highly efficient OPVs for the R2R slot-die coating process.

## 1. Introduction

As an emerging renewable energy source, bulk-heterojunction (BHJ) OPVs have attracted much attention due to the advantages of being lightweight, low manufacturing cost, high flexibility, and ease of large-area mass production [1,2,3,4]. Three key factors in the commercialization of OPVs include high PCE, long operating lifetime, and well-developed mass-production techniques. Many research groups have put a lot of effort on how to improve the PCE of OPVs. Previous studies mainly focus on the development of novel conducting polymers or small molecules as electron donor materials, such as poly[[4,8-bis[(2-ethylhexyl)oxy]benzo[1,2-b:4,5-b′]dithiophene-2,6-diyl][3-fluoro-2-[(2-ethylhexyl)carbonyl]thieno[3,4-b]thiophenediyl]] (PTB7), poly([2,6′-4,8-di(5-ethylhexylthienyl) benzo[1,2-b;3,3-b]dithiophene][3-fluoro-2[(2-ethylhexyl)carbonyl]thieno [3,4-b] thiophenediyl]) (PTB7-Th), Poly[(2,6-(4,8-bis(5-(2-ethylhexyl-3-fluoro)thiophen-2-yl)-benzo[1,2-b:4,5-b′]dithiophene))-alt-(5,5-(1′,3′-di-2-thienyl-5′,7′-bis(2-ethylhexyl)benzo[1′,2′-c:4′,5′-c′]dithiophene-4,8-dione)] (PM6) and 2,2′-[(3,3‴,3⁗,4′-tetraoctyl[2,2′:5′,2″:5″,2‴:5‴,2⁗-quinquethiophene]-5,5⁗-diyl)bis[(Z)-methylidyne(3-ethyl-4-oxo-5,2-thiazolidinediylidene)]]bis-propanedinitrile (DRCN5T) [5,6,7,8,9,10,11,12,13,14]. By incorporating interface engineering in device manufacturing, an improved PCE of over 18% has been achieved [15]. Although several new donor materials have been effectively used in high-efficiency OPVs, the electron acceptor materials are still limited to fullerene derivatives such as [6,6] phenyl-C61-butyric acid methyl ester (PC61BM), [6,6]-Phenyl-C71-butyric acid methyl ester (PC71BM) and 1′,1″,4′,4″-tetrahydro-di[1,4] methanonaphthaleno[5,6]fullerene-C60 (ICBA) for decades [10,16,17,18,19]. However, facilitating improvements in the PCE of OPVs was restricted by the weak absorption and lack of bandgap tunability of these fullerene derivatives. In addition to the PCE of OPVs, the fullerene aggregation during heating conditions often raises thermal stability concerns. In recent years, non-fullerene small molecule materials have emerged as alternative electron acceptor materials to fullerene derivatives [20,21,22,23,24,25]. Non-fullerene acceptor materials offer a significant advantage in their ability to be chemically modulated to fine-tune their absorption, energy level, and electronic mobility, which is challenging to achieve with fullerene derivatives. Various non-fullerene acceptors, including rylene imide, indacenodithiophene (IDT), and diketopyrrolopyrrole (DPP)-based acceptors, were broadly used to fabricate OPVs [26,27]. According to a previous study, single-junction OPVs based on low bandgap polymers and non-fullerene molecules achieve a PCE of about 18% [15], while the tandem structure of OPVs reaches over 19% [28]. However, these OPVs with promising PCEs were obtained by lab-scale spin coating processes, and the high-efficiency devices fabricated by the scalable and roll-to-roll (R2R) compatible method were still lacking.

Large-area solution-processable techniques have been successfully applied in OPV manufacturing, such as ultrasonic spray [29,30], inkjet printing [31,32,33], doctor blades [34,35,36,37], and slot-die coating [38,39,40,41,42]. Among these techniques, the slot-die coating process is the most promising candidate for mass production due to its high throughput, low material waste, low cost, and rapid manufacturing speed. However, the film formation of the slot-die process is different from the conventional spin-coating process. For the R2R slot-die process, the solidification of the wet film is complicated and critical. There are seldom studies demonstrating the R2R manufacture of OPVs based on polymer and non-fullerene acceptor materials. Liu et al. demonstrated a blade-coated OPV based on PTB7-Th and ITIC with high PCEs of 9.5% on a glass substrate and 7.6% on a flexible substrate, respectively [43]. Vak et al. presented flexible OPVs based on PBDB-T and ITIC with a notable PCE of 8.77% fabricated via batch slot-die process. On the other hand, a PCE of 7.11% was attained from a flexible device fabricated using the R2R slot-die coating process [44]. These results show that the non-fullerene acceptors and R2R slot-die coating process have great potential to improve the PCE performance of OPVs. Moreover, the efficiency gap between lab-scale spin-coated devices and R2R slot-die coated devices is needed to explore to further improve the performance of the OPV devices.

In our previous study, flexible OPVs fabricated via the R2R slot-die coating process with a high PCE of over 7% were demonstrated [45]. In this case, PTB7: PC_71_BM was R2R slot-die coated, controlling the amount of solvent additive and oven temperature to obtain a high PCE for the R2R flexible OPVs. According to these results, we presented a universal approach for fabricating low-cost, large-area, and environmentally friendly flexible organic photovoltaics. In this study, we present a new approach to fabricating R2R slot-die flexible OPVs utilizing low bandgap polymer (PBDB-T) and non-fullerene acceptor material (ITIC). In addition, we give an illustration of how the BHJ film structure and performance of the R2R slot-die coated non-fullerene-based OPV devices can be effectively improved by tuning the additive solvent, varying the slot-die coated parameters, and controlling the thermal processing conditions. Here, the flexible R2R slot-die coated PBDB-T/ITIC active layer notably achieved a PCE of 8.9%. This study paves the way to realize the commercialization of OPVs.

## 2. Materials and Methods

### 2.1. Preparation of Solution

The materials used were the same as our previous literature [45,46], and the ZnO nanoparticles (NPs) solution was based on our previous literature [47]. First, 4.4 g zinc acetate di-hydrate was dissolved in 220 mL ethanol until it completely dissolved. Then, 1.1 g lithium hydroxide monohydrate and 4 mL DI water were added to the previous solution, followed by vigorous stirring. After the color of the solution changed to transparent, the solution was put in a water bath at 60 °C and stirred for 30 min. The reacted solution was then centrifuged at 3000 rpm for 3 min, and then the suspension was removed. The ZnO powder was redispersed in IPA at a concentration of 10 mg/mL with 0.15% ethanolamine as a dispersant. For the active layer solution, 10 mg PBDB-T and 10 mg ITIC were dissolved in 1 mL chlorobenzene (CB) and stirred at 70 °C overnight before use.

### 2.2. Preparation of Devices

The electron transport layer (ETL) and photoactive layer were slot-die coated by the Coatema R2R system (Smartcoater, Coatema Coating Machinery GmbH, Dormagen, Germany). We use ITO-coated polyethylene terephthalate (PET) with a sheet resistance of 15 Ω/square from Optical Filters Ltd. (EMI ITO-15) as the flexible transparent conducting substrate. The PET/ITO substrate was first cut into a 10 × 10 cm^2^ sheet, and then the substrate was treated with air plasma. ZnO nanoparticles were used as the ETL in this study, and the PBDB-T:ITIC blend was used as the photoactive layer. The photoactive solutions with varying amounts of 1,8-diiodooctane (DIO) were stirred at 50 °C overnight before slot-die coating. The slot-die coated layer was dried immediately by an in-line oven equipped with an R2R-coating machine (R2R oven), which was illustrated in our previous literature [48]. The deposited ETL was dried at 150 °C for 10 min in an ambient oven, while the top deposited wet photoactive layer was dried under the R2R oven. The photoactive layers were subsequently post-annealed at 160 °C for 30 min in the ambient oven. To provide a hole transporting layer (HTL) and top metal electrode, MoO_3_ (5 nm) and then the Ag (100 nm) films were thermally evaporated on the photoactive layer by shadow mask. The devices used in this study have an area of 1 × 0.3 cm^2^. Figure 1 shows the chemical schematics of the donor and acceptor materials and the device structure. It is noteworthy to mention that all the R2R slot-die coating processes were conducted in ambient conditions.

### 2.3. Instrumentation

The current density-voltage characteristics of the devices were measured by a source meter (Keithley 2400, Keithley instruments, Cleveland, OH, U.S.) under AM 1.5G illumination (100 mW/cm^2^) using a solar simulator (Model #11000, Abet Technologies, Milford, CT, U.S.). The thicknesses of films were measured using a profilometer (Alpha Step D-100, KLA Tencor, Milpitas, CA, U.S.). The morphologies of R2R slot-die coated film were measured by atomic force microscopy (AFM, Innova, Bruker Corporation, Billerica, MA, U.S.). All the slot-die coated devices were not encapsulated and measured under ambient conditions.

## 3. Results and Discussion

High-performance OPVs based on PBDB-T and ITIC have been extensively reported in numerous literature [17,49,50,51]. Most of these studies demonstrate highly efficient devices that were typically fabricated under an inert atmosphere. To achieve future commercialization, we demonstrate an approach employing a slot-die process under ambient conditions. To evaluate how the processing environment affects OPV performance, we initially observed devices fabricated under different conditions. The current density-voltage (J-V) curves of these devices are shown in Figure 2a, with electrical characteristics provided in Table 1. In this study, the inverted structure of the devices was formed by spin-coating technique while keeping the weight ratio of PBDT-T and ITIC at 1:1. Here, the CB solution mixed with 0.5 vol% DIO was used as the processing solvent for fabricating the BHJ photoactive layer [49]. As illustrated in Figure 2a, the OPVs fabricated in a glove box filled with N_2_ show an open-circuit voltage (V_oc_) of 0.90 V, a short-circuit current density (J_sc_) of 17.68 mA/cm^2^, and a fill factor (FF) of 0.61, resulting in an average PCE of 9.75%, of which the highest PCE is 10.17%. OPVs fabricated under ambient conditions (with a relative humidity of ~50%) achieved an average PCE of 8.67%. The champion device exhibited a PCE of 8.88%, followed by a V_oc_ of 0.89 V, a J_sc_ of 15.72 mA/cm^2^, and a FF of 0.62. Interestingly, the PCE of devices produced in ambient conditions was approximately 10% lower than those manufactured under N_2_. In addition, the films deposited via the slot-die process exhibited a different thickness as compared to that via spin coating. This prompted us to investigate the effect of film thickness on the PCE of the devices based on the blend of PBDB-T:ITIC fabricated via the spin coating process. The related J-V curves and electrical characteristics are shown in Figure 2a and Table 1. Here, the effect of the thickness of the photoactive layer on the PCE by varying the deposition spin rate from 3000 rpm to 5000 rpm was observed. As demonstrated in using these parameters, the PCE was shown to further increase up to ~10% as the spin rate increases (lower film thickness). Based on the pre-testing results, the fabricating process was switched from spin coating to the R2R slot-die coating technique.

To optimize the R2R slot-die coating processing condition for the devices based on PBDB-T:ITIC, the parameters were focused on these processing parameters, including R2R oven temperature, DIO additive amount, post-annealing treatment condition, and R2R coating condition. Based on our previous research, we have developed a universal approach, which interrelates the effects of R2R oven temperature and additive amount to improve the PCE of the flexible OPVs [45]. Initially, the R2R oven temperature and the amount of DIO solvent additive were evaluated. The preliminary tests indicated the critical role of the R2R oven temperature for R2R slot-die coated flexible OPVs. In this study, the R2R oven temperatures were controlled to 100 °C, 120 °C, and 140 °C, which is consistent with our previous study [45]. The J-V curves of the devices fabricated via the R2R slot-die coating process with varying oven temperatures are shown in Figure 2b, while the performances of the devices are listed in Table 2. Here, the average PCE of the devices with an R2R oven temperature of 100 °C was found to be 6.78%, while the average PCEs of the devices were achieved at 7.15% as the R2R oven temperature increased to 120 °C. As the R2R oven temperature increased to 140 °C, the PCE of devices showed a drop of up to 6.70% which is suggested to be due to the J_SC_ and FF loss of devices. This result indicates that the fast solvent evaporation does not assist in the formation of the active layer at 140 °C. Therefore, the R2R process with an oven temperature of 120 °C was selected for further investigations.

To manipulate the film formation and improve the PCE, a high boiling-point DIO solvent additive was incorporated into the active layer solution as an additional and/or a new parameter. The effect of varying amounts of DIO additive (0, 0.25, 0.5, and 1 vol%) on the performance of the PBDB-T:ITIC active layer-based R2R-coated devices (R2R oven temperature is 120 °C) was observed. Figure 3a shows the J-V curves of these devices with the corresponding photovoltaic characteristics listed in Table 3. By adding 0.25 vol% DIO in the photoactive solution, the J_sc_ of R2R slot-die OPVs improved significantly, resulting in an enhanced PCE of ~8%. With a further increase in the DIO amount from 0.5 vol% to 1 vol%, the PCE of R2R slot-die OPVs was decreased to 7.1%, where the PCE reduction is due to the lower J_sc_ and FF. It is noteworthy that the R2R slot-die coated photoactive layer was not completely dried when passing through the R2R oven at 120 °C due to the high boiling point of the DIO additive. The wet photoactive layer was irregularly deposited on the substrate as it proceeded through the roller. To evaluate the DIO’s effect on the PCE’s performance, we use AFM to measure the roughness of the photoactive layer, shown in Figure 4. Compared to the photoactive layer without DIO, the roughness of the photoactive layer with 0.25 vol% DIO was reduced from 1.82 nm to 0.98 nm, which is expected to correspond to the performance of the device. After the amount of the DIO was increased to 1 vol%, the roughness of the active layer was slightly increased in comparison to that with 0.5 vol% DIO. It is suggested that excess DIO additive on the active layer demonstrates a negative impact on the photovoltaic performance of OPV. Further, PBDB−T:ITIC: 0.25 vol% DIO film was R2R slot-die coated in the substrate and post-thermally annealed with varying thermally annealing times of 10, 20, and 30 min. The results of the related J-V curves and electric characteristics of the samples are shown in Figure 3b and Table 4. Post-thermal annealing of 140 °C with 30 min displayed the highest FF, resulting in a notable PCE of 8.70%. The impact of annealing temperature on device performance is detailed in Table 5, where all samples were subjected to post-annealing for a consistent duration of 30 min. The average PCE of OPVs based on the active layer annealed at 120 °C does not show an obvious improvement compared to the OPV without post-annealing. By increasing the annealing temperature to 140 °C, the average PCE of OPVs can be enhanced from 8.10% to 8.60%. It indicated that the active layer with DIO additive was through sufficient recrystallization at 140 °C. Conversely, a decline in PCE was observed when the annealing temperature was further increased to 160 °C. It speculated that the active layer suffered from the phase segregation of PBDB-T during the post-annealing process [51]. Therefore, it is hypothesized that extending the annealing time beyond 30 min could be detrimental to OPV performance, primarily due to the potential phase segregation of the active layer. To maintain optimal blend homogeneity in the active layer, we standardized the annealing time to 30 min.

The performance of OPVs is significantly influenced by the quality of the active layer, which is determined by the R2R coating condition. We further investigate the effect of R2R slot-die coating parameters, such as ink injection rate and coating speed, on the morphology, uniformity, and thickness of the photoactive layer. The resulting J-V curves and performance of OPVs (PBDB-T:ITIC: 0.25 vol% DIO) with varying coating conditions are shown in Figure 5a and Table 6. Following the optimized R2R slot-die coating parameters, a prominent PCE of over 9% is achieved. Figure 5b presents the PCEs of the OPV device with varying film thicknesses. The thickness of the active layer was controlled by the coating parameters, such as input rate and coating speed. In the case using 0.5 mL·min^−1^ of input rate, the PCE of OPVs can be improved when the coating speed is increased from 0.7 to 1.0 m·min^−1^. It implied faster coating speed may result in more uniform film. Nevertheless, these devices exhibited a low J_sc_ due to the inadequate absorption of the active layer. To obtain the thick active layer, we increase the input rate to 0.7 mL·min^−1^. Under this condition, raising the coating speed from 0.7 to 1.0 m·min^−1^, the PCE of OPVs can be enhanced significantly from 7.2% to 8.6%. With the coating speed of 1.0 m·min^−1^, the thickness of the active layer was increased to 170 nm using 1.0 mL·min^−1^ of input rate. However, this led to a decline in PCE, suggesting more bulk defects in the thicker photoactive layer. To optimize the quality of the active layer coating, the speed was adjusted to 1.4 m·min^−1^, which resulted in OPVs exhibiting a commendable PCE of 8.9%. Figure 6 shows the AFM images of the R2R slot-die coated films via various coating parameters. We can observe that the surface morphology is influenced by the variation in the R2R slot-die coating parameters. The smoothest surface of the photoactive layer was obtained at 1.0/1.4 with a film thickness of 120 nm. As the thickness of the active layer increased to 170 nm, the roughness of the active layer increased from 0.84 nm to 5.28 nm. This represents the criticality of precise R2R slot-die coating parameters for ensuring high-quality films.

To demonstrate the large-area capability of R2R slot-die coated OPVs, we further evaluated the large-area R2R OPVs devices and modules. Figure 7a and Table 7 show the performance of OPVs with different device areas. The OPV with an active area of 0.3 cm^2^ displayed the highest PCE of 6.90%. It can be observed that as the active area is increased to 4 cm^2^, the PCE of the OPVs decreases to 5.70% owing to the FF loss. It implied that carrier recombination was dominating during long-distance transporting [52]. Moreover, we demonstrated an R2R OPV module, which has eight interconnecting sub-cells, and the J-V curve of the R2R OPV module is shown in Figure 7b. The total active area of the module was 3.2 cm^2^, and the PCE of the module was achieved up to 6.30%.

## 4. Conclusions

We developed a facile R2R slot-die coating approach to significantly enhance the PCE of flexible OPVs based on the PBDB-T:ITIC blends. By adjusting the DIO solvent additive, R2R oven temperature, post-thermal annealing conditions, and module design, we showcased the capabilities of the R2R slot-die coated OPVs. The amount of DIO additive in the photoactive layer solution was demonstrated to significantly improve the performance of the OPV by tailoring the morphology and intermixing phase separation of the R2R coating active layer. A notable PCE of 8.9% based on the R2R flexible OPV device was achieved by optimizing the R2R slot-die coating process. Our study demonstrates the feasibility of solution-processable slot-die coated high-performance OPVs.

## Figures and Tables

**Figure 1 polymers-15-04005-f001:**
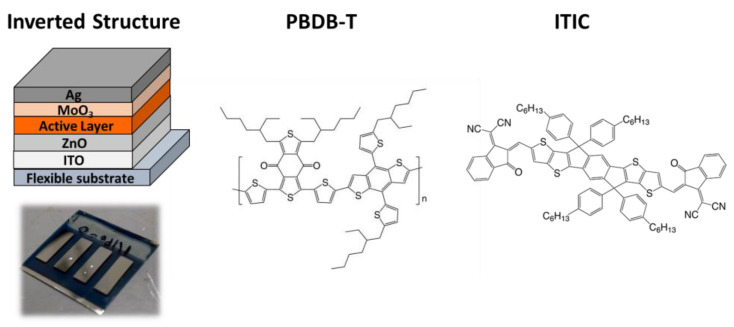
The structure of the R2R slot-die coated devices and chemical structure of PBDB-T and ITIC.

**Figure 2 polymers-15-04005-f002:**
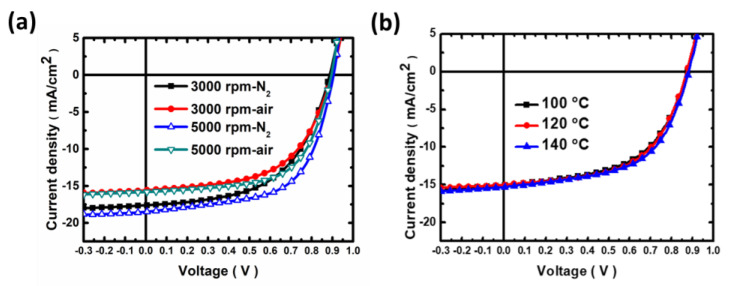
(**a**) J-V curves of OPVs based on PBDB−T:ITIC fabricated via the spin coating process with varying spin rates and atmospheric conditions. (**b**) J-V curves of OPVs based on PBDB−T:ITIC fabricated using the R2R slot-die coating process with various R2R oven temperatures.

**Figure 3 polymers-15-04005-f003:**
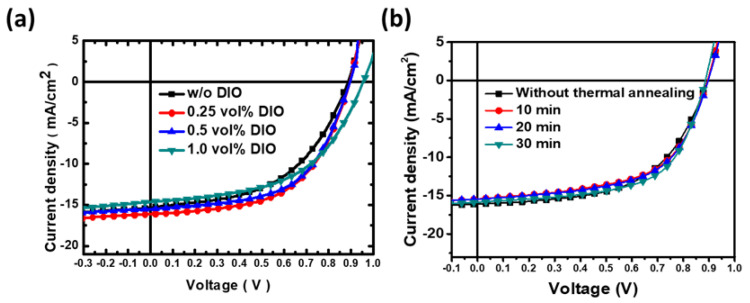
(**a**) J−V curves of OPVs based on PBDB−T:ITIC:DOI with varying amounts of DIO additive prepared via the R2R slot-die coating process under 120 °C of R2R oven temperature. (**b**) J−V curves of OPVs based on PBDB−T:ITIC fabricated via R2R slot-die coating process under 120 °C oven temperature with post-thermal annealing temperature of 160 °C with varying annealing times.

**Figure 4 polymers-15-04005-f004:**
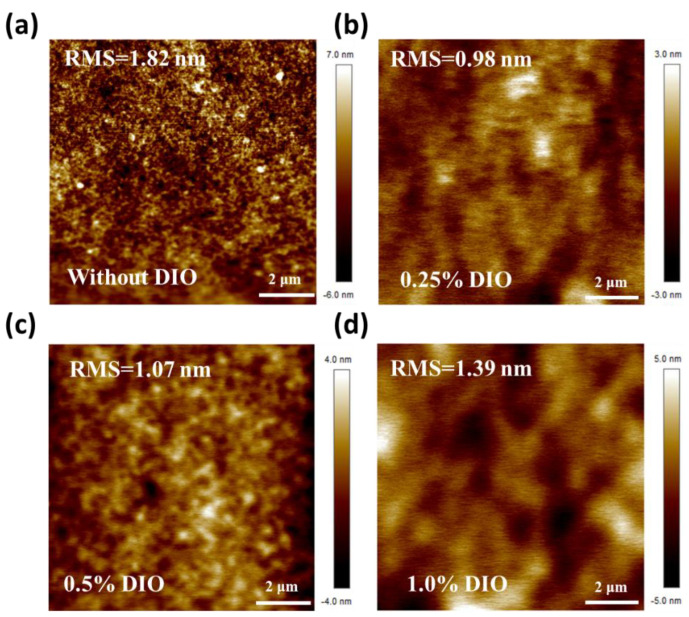
AFM images of OPVs based on PBDB−T:ITIC with varying amounts of DIO additive fabricated via the R2R slot-die coating process under 120 °C oven temperature: (**a**) Without DIO, (**b**) 0.25 vol% DIO, (**c**) 0.5 vol% DIO, (**d**) 1.0 vol% DIO.

**Figure 5 polymers-15-04005-f005:**
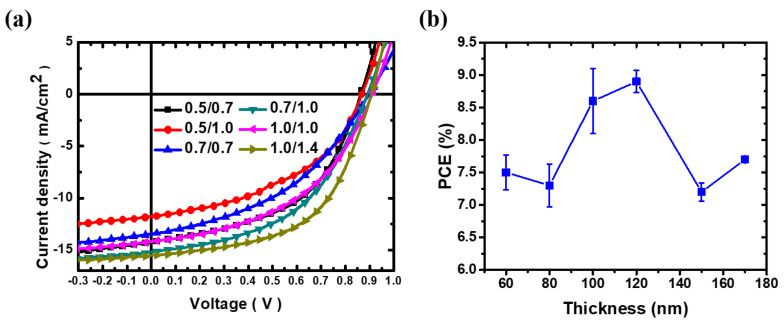
(**a**) J-V curves of the devices based on PBDB−T:ITIC:0.25 vol% DIO, (**b**) plot of the thickness of the active layer versus PCE of OPV fabricated via the R2R slot-die coating process with various coating conditions under 120 °C R2R oven temperature.

**Figure 6 polymers-15-04005-f006:**
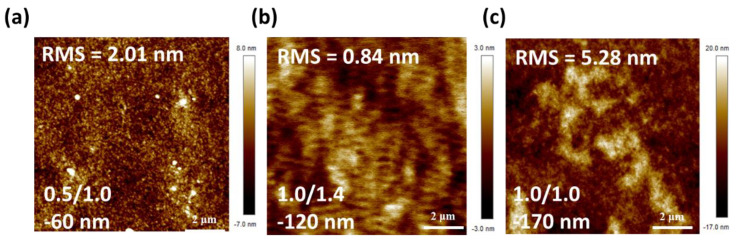
AFM images of PBDB−T:ITIC fabricated via the R2R slot-die coating process with various coating conditions under 120 °C R2R oven temperature. (**a**) 0.5/1.0, (**b**) 1.0/1.4, (**c**) 1.0/1.0.

**Figure 7 polymers-15-04005-f007:**
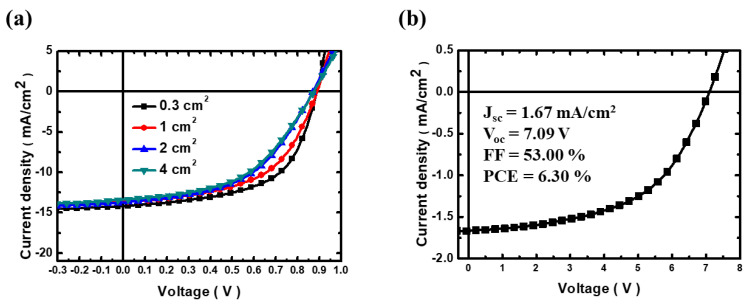
(**a**) J−V curves of R2R OPVs with various areas of the active layer, (**b**) J−V curve of R2R OPV module interconnected in 8 sub-cells.

**Table 1 polymers-15-04005-t001:** Photovoltaic characteristics of OPVs based on an PBDB−T:ITIC active layer prepared via spin coating with various film thicknesses under different processing atmospheres. (These values are for the champion PCE of these devices, and average values are obtained from 15 devices in the brackets).

Spin Rate (rpm)	Atmosphere	J_sc_ (mA/cm^2^)	V_oc_ (V)	FF	PCE (%)
3000	N_2_	17.62 (17.66 ± 0.24)	0.89 (0.89 ± 0.00)	0.55 (0.53 ± 0.011)	8.62 (8.40 ± 0.16)
	Air	15.58 (15.58 ± 0.14)	0.88 (0.89 ± 0.01)	0.59 (0.57 ± 0.014)	8.09 (7.93 ± 0.24)
5000	N_2_	18.52 (17.68 ± 0.75)	0.90 (0.90 ± 0.01)	0.61 (0.61 ± 0.011)	10.17 (9.75 ± 0.30)
	Air	15.83 (15.72 ± 0.10)	0.89 (0.89 ± 0.01)	0.63 (0.62 ± 0.006)	8.88 (8.67 ± 0.12)

**Table 2 polymers-15-04005-t002:** Photovoltaic characteristics of OPVs based on PBDB−T:ITIC prepared via the R2R slot-die coating process under varying R2R oven temperatures. These values are for the highest PCE of these devices, and the average values are obtained from 15 devices in the brackets.

R2R Oven Temp. (°C)	J_sc_ (mA/cm^2^)	V_oc_ (V)	FF	PCE (%)
100	14.63 (14.71 ± 0.15)	0.86 (0.86 ± 0.01)	0.56 (0.54 ± 0.03)	7.05 (6.78 ± 0.33)
120	14.80 (14.70 ± 0.26)	0.87 (0.86 ± 0.01)	0.57 (0.56 ± 0.01)	7.34 (7.15 ± 0.10)
140	14.42 (14.50 ± 0.07)	0.86 (0.86 ± 0.01)	0.54 (0.53 ± 0.01)	6.70 (6.60 ± 0.08)

**Table 3 polymers-15-04005-t003:** Photovoltaic characteristics of OPVs based on PBDB−T:ITIC:DIO with varying amounts of DIO additive prepared via the R2R slot-die coating process under 120 °C oven temperature. These values represent the champion PCE of these devices, and the average data are obtained from 20 devices in the brackets.

DIO Amount (vol%)	J_sc_ (mA/cm^2^)	V_oc_ (V)	FF	PCE (%)
w/o	15.30 (15.29 ± 0.05)	0.86 (0.86 ± 0.01)	0.53 (0.52 ± 0.01)	7.00 (6.93 ± 0.06)
0.25	16.22 (16.12 ± 0.11)	0.90 (0.90 ± 0.01)	0.56 (0.56 ± 0.01)	8.20 (8.07 ± 0.15)
0.5	15.44 (15.69 ± 0.38)	0.90 (0.90 ± 0.01)	0.57 (0.56 ± 0.01)	7.90 (7.90 ± 0.03)
1	14.65 (14.24 ± 0.36)	0.96 (0.94 ± 0.01)	0.53 (0.53 ± 0.01)	7.30 (7.10 ± 0.17)

**Table 4 polymers-15-04005-t004:** Photovoltaic characteristics of OPVs based on PBDB-T:ITIC fabricated via the R2R slot-die coating process under 120 °C oven temperature with post-thermal annealing temperature of 140 °C with varying annealing times. These values are for the champion PCE of the devices, and the average data in the brackets are obtained from 20 devices.

Annealing Time (min)	J_sc_ (mA/cm^2^)	V_oc_ (V)	FF	PCE (%)
w/o	16.23 (16.41 ± 0.40)	0.90 (0.89 ± 0.01)	0.56 (0.56 ± 0.01)	8.20 (8.10 ± 0.10)
10	15.47 (15.46 ± 0.05)	0.90 (0.89 ± 0.01)	0.58 (0.59 ± 0.01)	8.10 (8.07 ± 0.06)
20	15.54 (15.50 ± 0.02)	0.89 (0.89 ± 0.01)	0.59 (0.59 ± 0.01)	8.20 (8.16 ± 0.06)
30	16.04 (15.81 ± 0.20)	0.89 (0.89 ± 0.01)	0.60 (0.61 ± 0.01)	8.70 (8.60 ± 0.10)

**Table 5 polymers-15-04005-t005:** Photovoltaic characteristics of OPVs based on PBDB−T:ITIC fabricated via the R2R slot-die coating process under 120 °C oven temperature with various post-thermal annealing temperatures for 30 min. These values are for the champion PCE of the devices, and the average data in the brackets are obtained from 20 devices.

Annealing Temperature (°C)	J_sc_ (mA/cm^2^)	V_oc_ (V)	FF	PCE (%)
w/o	16.23 (16.41 ± 0.40)	0.90 (0.89 ± 0.01)	0.56 (0.56 ± 0.01)	8.20 (8.10 ± 0.10)
120	16.07 (15.91 ± 0.15)	0.90 (0.87 ± 0.04)	0.60 (0.58 ± 0.03)	8.61 (8.06 ± 0.51)
140	16.04 (15.81 ± 0.20)	0.89 (0.89 ± 0.01)	0.60 (0.61 ± 0.01)	8.70 (8.60 ± 0.10)
160	14.10 (14.29 ± 0.04)	0.86 (0.82 ± 0.01)	0.55 (0.57 ± 0.01)	6.70 (6.63 ± 0.09)

**Table 6 polymers-15-04005-t006:** Performance of OPVs based on the active layer consisting of PBDB−T:ITIC:0.25 vol% DIO fabricated via the R2R slot-die coating process with various coating conditions under 120 °C R2R oven temperature. These values represent the highest PCE of the devices, and the average data are obtained from 20 devices in the brackets.

Input Rate/ Coating Speed (mL·min^−1^/m·min^−1^)	J_sc_ (mA/cm^2^)	V_oc_ (V)	FF	PCE (%)	Thickness (nm)
0.5/0.7	14.50 (14.60 ± 0.31)	0.88 (0.88 ± 0.01)	0.58 (0.56 ± 0.01)	7.30 (7.22 ± 0.33)	80.6 ± 1.5
0.5/1.0	13.80 (13.71 ± 0.11)	0.90 (0.90 ± 0.01)	0.60 (0.59 ± 0.02)	7.50 (7.30 ± 0.27)	59.6 ± 0.9
0.7/0.7	15.13 (15.13 ± 0.01)	0.89 (0.88 ± 0.01)	0.54 (0.53 ± 0.06)	7.20 (7.10 ± 0.14)	150.2 ± 3.3
0.7/1.0	16.08 (15.92 ± 0.15)	0.91 (0.88 ± 0.05)	0.59 (0.58 ± 0.04)	8.60 (8.07 ± 0.50)	100.2 ± 1.1
1.0/1.0	16.68 (16.71 ± 0.05)	0.89 (0.89 ± 0.01)	0.52 (0.52 ± 0.01)	7.70 (7.70 ± 0.01)	170.6 ± 2.2
1.0/1.4	17.42 (17.39 ± 0.44)	0.90 (0.90 ± 0.01)	0.57 (0.55 ± 0.01)	8.90 (8.75 ± 0.17)	120.1 ± 2.0

**Table 7 polymers-15-04005-t007:** Performance of R2R OPVs with different areas of active layers.

Active Area (cm^2^)	J_sc_ (mA/cm^2^)	V_oc_ (V)	FF	PCE (%)
1 × 0.3	14.29	0.88	54.68	6.90
1 × 1	14.21	0.88	50.30	6.30
1 × 2	13.88	0.89	46.90	5.80
1 × 4	13.66	0.88	47.10	5.70

## Data Availability

The data presented in this study are available on request from the corresponding authors.

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
