# Peer review of "Highly Efficient Flexible Roll-to-Roll Organic Photovoltaics Based on Non-Fullerene Acceptors"

_polymers, 2023, doi:10.3390/polym15194005_

Round 1
Reviewer 1 Report
See attached

Author Response
Dear Reviewer,
Thanks for your valuable comments and suggestions. We have replied to all comments accordingly per the attached file.

Reviewer 2 Report
Dear authors,
The PBDB-T: ITIC based organic photovoltaic device is fabricated, wherein the active layer is deposited via R2R approach. The materials characterization and the obtained device efficiency is reasonable to appreciate with. The listed comments may be considered to further improve the visibility of the work.
Minor comments
Abstract
(1) Line 16: Avoid the use of ‘flexible OPVs’ twice in the same line!
(2) Line 19: Expand ‘PBDB-T’ and ‘ITIC’ before their abbreviation.
Experimental
(1) Line 107: Should it be ‘sheet resistance’ instead of ‘surface resistance’?
Results and discussion
(1) How about the thickness dependent performance of the active layer?

Author Response

(The authors gave the same response as above.)

Reviewer 3 Report
This study studies flexible organic photovoltaics (OPVs) fabricated via roll-to-roll (R2R) processing and underscores the importance of optimizing R2R processing parameters in order to achieve optimal performance. This study aligns with Polymers’ scope but it lacks originality as it is very similar to a prior publication. The manuscript itself was also poorly written. There are instances of awkward sentence structures and grammatical errors throughout the document. Overall, I don’t recommend this paper being published. Please see my comments below.
Major comments
Lack of novelty and originality
This paper feels like a replicate of a paper by Na et al (https://doi.org/10.1002/adfm.201805825). Both papers used an R2R slot die coating process to manufacture devices using PBDB-T:ITIC. Although this paper showed how different DIO vol% would impact the device performance and briefly discussed how PCE deteriorates with the size of OPV device, both phenomena have already been well-known to the OPV community for years. The authors were also able to achieve some performance improvement by adding DIO, it was not fundamentally different from what was reported by Na et al. and did not make any meaningful contribution to solving the commercialization challenges of OPV (e.g., stability, scalability).
Poor writing quality
The manuscript itself is not well-written. I found numerous grammar mistakes and poorly constructed sentences throughout the article. I have pointed out quite a few in the detailed comments section below. I recommend a thorough proofreading and editing process to address these issues. The authors may consider involving a professional editor or using grammar-checking software.
Detailed comments (in line number)
17: replace “roll-to-roll” with “R2R”
21: remove details “(AM 1.5G, 100 mW/cm2)”
22-24: please improve the language. It is essentially “we demonstrated results by adjusting parameters” which feels very awkward
30,32,64: no need to redefine OPV / PCE / R2R as they are already defined in Abstract
36-44: please make sure to cite the articles that first reported each material here. For example, PTB7 was first reported by Liang et al. (https://doi.org/10.1002/adma.200903528) and it is not cited here
61-62: please use more active voice and less passive voice
100: “similar” is not the same. If it’s different from the reported procedure, please elaborate to ensure the reproducibility
118: can the authors add a picture of the actual device?
138: please improve readability
148-149: please improve readability
237 / Figure 5b: The PCE for devices with a 60 and 170nm photoactive layer are unusual. I would expect them to have lower PCE compared to their neighbors. Please explain what led to the higher PCE
238-239: please improve readability
266-267: please share evidence that supports the claim
277-279: please improve readability
The writing quality is poor. Please see comments and suggestions for details
Author Response
Dear Reviewer,
Thanks for your valuable comments and suggestions. We have replied to all the comments per the attached file.

Round 2
Reviewer 3 Report
I'm glad to see that the authors have significantly improved the quality of their manuscript and I don't see any glaring issue right now. I recommend accepting the manuscript as is but notice that there is still some room for improvement regarding languages / writing styles. Please see the language section for details
Overall good now. I encourage the authors to use more active voice. Some examples:
Ln47-49: can be "...Liu et al. achieved an improved PCE of..."
Ln85-86: can be "we demonstrated..."